# STOCHASTIC VARIATIONAL VIDEO PREDICTION

Mohammad Babaeizadeh[1], Chelsea Finn[2], Dumitru Erhan[3], Roy Campbell[1], and Sergey Levine[2,3]

[1]University of Illinois at Urbana-Champaign
[2]University of California, Berkeley
[3]Google Brain

mb2@uiuc.edu, cbfinn@eecs.berkeley.edu, dumitru@google.com,
rhc@illinois.edu, svlevine@eecs.berkeley.edu

## ABSTRACT

Predicting the future in real-world settings, particularly from raw sensory observations such as images, is exceptionally challenging. Real-world events can be stochastic and unpredictable, and the high dimensionality and complexity of natural images require the predictive model to build an intricate understanding of the natural world. Many existing methods tackle this problem by making simplifying assumptions about the environment. One common assumption is that the outcome is deterministic and there is only one plausible future. This can lead to low-quality predictions in real-world settings with stochastic dynamics. In this paper, we develop a stochastic variational video prediction (SV2P) method that predicts a different possible future for each sample of its latent variables. To the best of our knowledge, our model is the first to provide effective stochastic multi-frame prediction for real-world videos. We demonstrate the capability of the proposed method in predicting detailed future frames of videos on multiple real-world datasets, both action-free and action-conditioned. We find that our proposed method produces substantially improved video predictions when compared to the same model without stochasticity, and to other stochastic video prediction methods. Our SV2P implementation will be open sourced upon publication.

## 1 INTRODUCTION

Understanding the interaction dynamics of objects and predicting what happens next is one of the key capabilities of humans which we heavily rely on to make decisions in everyday life (Bubic et al., 2010). A model that can accurately predict future observations of complex sensory modalities such as vision must internally represent the complex dynamics of real-world objects and people, and therefore is more likely to acquire a representation that can be used for a variety of visual perception tasks, such as object tracking and action recognition (Srivastava et al., 2015; Lotter et al., 2017; Denton & Birodkar, 2017). Furthermore, such models can be inherently useful themselves, for example, to allow an autonomous agent or robot to decide how to interact with the world to bring about a desired outcome (Oh et al., 2015; Finn & Levine, 2017).

However, modeling future distributions over images is a challenging task, given the high dimensionality of the data and the complex dynamics of the environment. Hence, it is common to make various simplifying assumptions. One particularly common assumption is that the environment is deterministic and that there is only one possible future (Chiappa et al., 2017; Srivastava et al., 2015; Boots et al., 2014; Lotter et al., 2017). Models conditioned on the actions of an agent frequently make this assumption, since the world is more deterministic in these settings (Oh et al., 2015; Finn et al., 2016). However, most real-world prediction tasks, including the action-conditioned settings, are in fact not deterministic, and a deterministic model can lose many of the nuances that are present in real physical interactions. Given the stochastic nature of video prediction, any deterministic model is obliged to predict a *statistic* of all the possible outcomes. For example, deterministic models trained with a mean squared error loss function generate the expected value of all the possibilities for each pixel independently, which is inherently blurry (Mathieu et al., 2016).

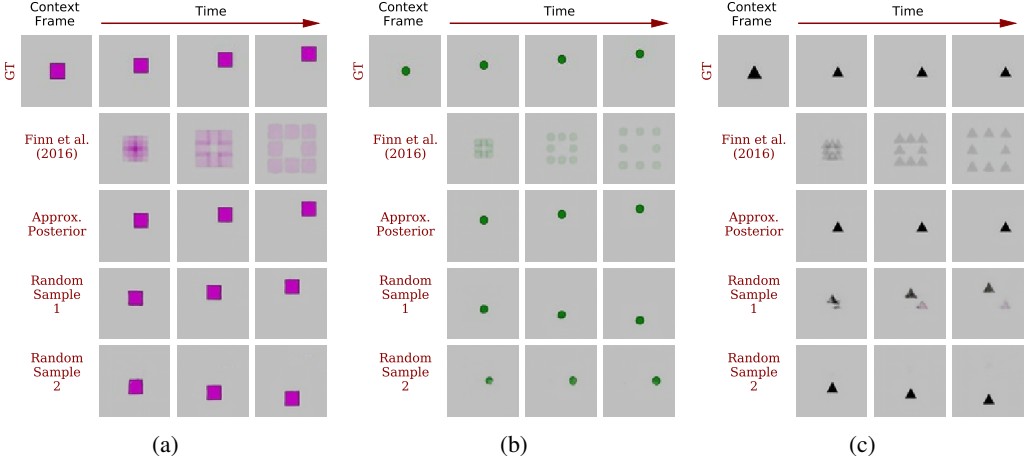

Figure 1: Importance of stochasticity in video prediction. In each video, a random shape follows a random direction (first row). Given only the first frame, the deterministic model from Finn et al. (2016) predicts the average of all the possibilities. The third row is the output of SV2P with latent sampled from approximated posterior which predicts the correct motion. Last two rows are stochastic outcomes using random latent values sampled from assumed prior. As observed, these outcomes are random but within the range of possible futures. Second sample of Figure 1c shows a case where the model predicts the average of more than one outcome.

Our main contribution in this paper is a stochastic variational method for video prediction, named SV2P, that predicts a different plausible future for each sample of its latent random variables. We also provide a stable training procedure for training a neural network based implementation of this method. To the extent of our knowledge, SV2P is the first latent variable model to successfully predict multiple frames in real-world settings. Our model also supports action-conditioned predictions, while still being able to predict stochastic outcomes of ambiguous actions, as exemplified in our experiments. We evaluate SV2P on multiple real-world video datasets, as well as a carefully designed toy dataset that highlights the importance of stochasticity in video prediction (see Figure 1). In both our qualitative and quantitative comparisons, SV2P produces substantially improved video predictions when compared to the same model without stochasticity, with respect to standard metrics such as PSNR and SSIM. The stochastic nature of SV2P is most apparent when viewing the predicted videos. Therefore, we highly encourage the reader to check the project website `https://goo.gl/iywUHc` to view the actual videos of the experiments. The Tensor-Flow (Abadi et al., 2016) implementation of this project will be open sourced upon publication.

## 2 RELATED WORK

A number of prior works have addressed video frame prediction while assuming deterministic environments (Ranzato et al., 2014; Srivastava et al., 2015; Vondrick et al., 2015; Xingjian et al., 2015; Boots et al., 2014; Lotter et al., 2017). In this work, we build on the deterministic video prediction model proposed by Finn et al. (2016), which generates the future frames by predicting the motion flow of dynamically masked out objects extracted from the previous frames. Similar transformation-based models were also proposed De Brabandere et al. (2016); Liu et al. (2017). Prior work has also considered alternative objectives for deterministic video prediction models to mitigate the blurriness of the predicted frames and produce sharper predictions (Mathieu et al., 2016; Vondrick & Torralba, 2017). Despite the adversarial objective, Mathieu et al. (2016) found that injecting noise did not lead to stochastic predictions, even for predicting a single frame. Oh et al. (2015); Chiappa et al. (2017) make sharp video predictions by assuming deterministic outcomes in video games given the actions of the agents. However, this assumption does not hold in real-world settings, which almost always have stochastic dynamics.

Auto-regressive models have been proposed for modeling the joint distribution of the raw pixels (Kalchbrenner et al., 2017). Although these models predict sharp images of the future, their

training and inference time is extremely high, making them difficult to use in practice. Reed et al. (2017) proposed a parallelized multi-scale algorithm that significantly improves the training and prediction time but still requires more than a minute to generate one second of $64 \times 64$ video on a GPU. Our comparisons suggest that the predictions from these models are sharp, but noisy, and that our method produces substantially better predictions, especially for longer horizons.

Another approach for stochastic prediction uses generative adversarial networks (GANs) (Goodfellow et al., 2014), which have been used for video generation and prediction (Tulyakov et al., 2017; Li et al., 2017). Vondrick et al. (2016); Chen et al. (2017) applied adversarial training to predict video from a single image. Although GANs generate sharp images, they tend to suffer from mode-collapse (Goodfellow, 2016), particularly in conditional generation settings (Zhu et al., 2017).

Variational auto-encoders (VAEs) (Kingma & Welling, 2014) also have been explored for stochastic prediction tasks. Walker et al. (2016) uses conditional VAEs to predict dense trajectories from pixels. Xue et al. (2016) predicts a single stochastic frame using cross convolutional networks in a VAE-like architecture. Shu et al. (2016) uses conditional VAEs and Gaussian mixture priors for stochastic prediction. Both of these works have been evaluated solely on synthetic datasets with simple moving sprites and no object interaction. Real images significantly complicate video prediction due to the diversity and variety of stochastic events that can occur. Fragkiadaki et al. (2017) compared various architectures for multimodal motion forecasting and one-frame video prediction, including variational inference and straightforward sampling from the prior. Unlike these prior models, our focus is on designing a multi-frame video prediction model to produce stochastic predictions of the future. Multi-frame prediction is dramatically harder than single-frame prediction, since complex events such as collisions require multiple frames to fully resolve, and single-frame predictions can simply ignore this complexity. We believe, our approach is the first latent variable model to successfully demonstrate stochastic multi-frame video prediction on real world datasets.

# 3 STOCHASTIC VARIATIONAL VIDEO PREDICTION (SV2P)

In order to construct our stochastic variational video prediction model, we first formulate a probabilistic graphical model that explains the stochasticity in the video. Since our goal is to perform conditional video prediction, the predictions are conditioned on a set of $c$ *context* frames $\mathbf{x}_0, \ldots, \mathbf{x}_{c-1}$ (e.g., if we are conditioning on one frame, $c = 1$), and our goal is to sample from $p(\mathbf{x}_{c:T}|\mathbf{x}_{0:c-1})$, where $\mathbf{x}_i$ denotes the i$^{\text{th}}$ frame of the video (Figure 2).

Video prediction is stochastic as a consequence of the latent events that are not observable from the context frames alone. For example, when a robot's arm pushes a toy on a table, the unknown weight of that toy affects how it moves. We therefore introduce a vector of latent variables $\mathbf{z}$ into our model, distributed according to a prior $\mathbf{z} \sim p(\mathbf{z})$, and build a model $p(\mathbf{x}_{c:T}|\mathbf{x}_{0:c-1}, \mathbf{z})$. This model is still stochastic but uses a more general representation, such as a conditional Gaussian, to explain just the noise in the image, while $\mathbf{z}$ accounts for the more complex stochastic phenomena. We can then factorize this model to $\prod_{t=c}^{T} p_\theta(\mathbf{x}_t|\mathbf{x}_{0:t-1}, \mathbf{z})$. Learning then involves training the parameters of these factors $\theta$, which we assume to be shared between all the time steps.

At inference time we need to estimate values for the true posterior $p(\mathbf{z}|\mathbf{x}_{0:T})$, which is intractable due its dependency on $p(\mathbf{x}_{0:T})$. We

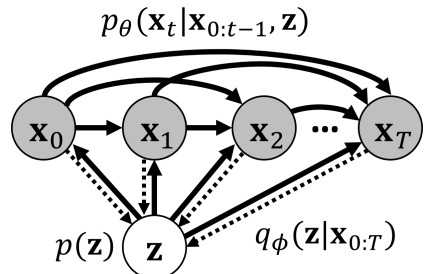

Figure 2: Probabilistic graphical model of stochastic variational video prediction, assuming time-invariant latent. The generative model predicts the next frame conditioned on the previous frames and latent variables (solid lines), while the variational inference model approximates the posterior given all the frames (dotted lines).

overcome this problem by approximating the posterior with an inference network $q_\phi(\mathbf{z}|\mathbf{x}_{0:T})$ that outputs the parameters of a conditionally Gaussian distribution $\mathcal{N}(\mu_\phi(\mathbf{x}_{0:T}), \sigma_\phi(\mathbf{x}_{0:T}))$. This net-

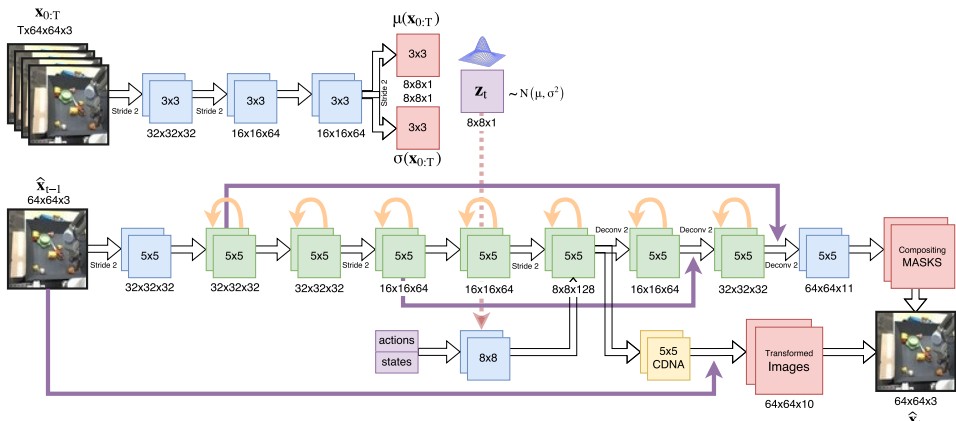

Figure 3: Architecture of SV2P. At training time, the inference network (top) estimates the posterior $q_\phi(\mathbf{z}|\mathbf{x}_{0:T}) = \mathcal{N}\big(\mu(\mathbf{x}_{0:T}), \sigma(\mathbf{x}_{0:T})\big)$. The latent value $\mathbf{z} \sim q_\phi(\mathbf{z}|\mathbf{x}_{0:T})$ is passed to the generative network along with the (optional) action. The generative network (from Finn et al. (2016)) predicts the next frame given the previous frames, latent values, and actions. At test time, $\mathbf{z}$ is sampled from the assumed prior $\mathcal{N}(\mathbf{0}, \mathbf{I})$.

work is trained using the reparameterization trick (Kingma & Welling, 2014), according to:

$$\mathbf{z} = \mu_\phi(\mathbf{x}_{0:T}) + \sigma_\phi(\mathbf{x}_{0:T}) \times \epsilon, \qquad \epsilon \sim \mathcal{N}(\mathbf{0}, \mathbf{I}) \tag{1}$$

Here, $\theta$ and $\phi$ are the parameters of the generative model and inference network, respectively. To learn these parameters, we can optimize the variational lower bound, as in the variational autoencoder (VAE) (Kingma & Welling, 2014):

$$\mathcal{L}(\mathbf{x}) = -\mathbb{E}_{q_\phi(\mathbf{z}|\mathbf{x}_{0:T})}\big[\log p_\theta(\mathbf{x}_{t:T}|\mathbf{x}_{0:t-1}, \mathbf{z})\big] + D_{KL}\big(q_\phi(\mathbf{z}|\mathbf{x}_{0:T})||p(\mathbf{z})\big) \tag{2}$$

where $D_{KL}$ is the Kullback-Leibler divergence between the approximated posterior and assumed prior $p(\mathbf{z})$ which in our case is the standard Gaussian $\mathcal{N}(\mathbf{0}, \mathbf{I})$.

In Equation 2, the first term on the RHS represents the reconstruction loss while the second term represents the divergence of the variational posterior from the prior on the latent variable. It is important to emphasize that the approximated posterior is conditioned on **all** of the frames, including the future frames $\mathbf{x}_{t:T}$. This is feasible during training, since $\mathbf{x}_{t:T}$ is available at the training time, while at test time we can sample the latents from the assumed prior. Since the aim in our method is to recover latent variables that correspond to events which might explain the variability in the videos, we found that it is in fact crucial to condition the inference network on future frames. At test time, the latent variables are simply sampled from the prior which corresponds to a smoothing-like inference process. In principle, we could also perform a filtering-like inference procedure of the form $q_\phi(\mathbf{z}|\mathbf{x}_{0:t-1})$ for time step $t$ to infer the most likely latent variables based only on the conditioning frames, instead of sampling from the prior, which could produce more accurate predictions at test time. However, it would be undesirable to use a filtering process at training time: in order to incentivize the forward prediction network to make use of the latent variables, they must contain some information that is useful for predicting future frames that is not already present in the context frames. If they are predicted entirely from the context frames, no such information is present, and indeed we found that a purely filtering-based model simply ignores the latent variables.

So far, we've assumed that the latent events are constant over the entire video. We can relax this assumption by conditioning prediction on a time-variant latent variable $\mathbf{z}_t$ that is sampled at every time step from $p(\mathbf{z})$. The generative model then becomes $p(\mathbf{z}_t) \prod_{t=c}^{T} p_\theta(\mathbf{x}_t|\mathbf{x}_{0:t-1}, \mathbf{z}_t)$ and, assuming a fixed posterior, the inference model will be approximated by $q_\phi(\mathbf{z}_t|\mathbf{x}_{0:T})$, where the model parameters $\phi$ are shared across time. In practice, the only difference between these two formulations is the frequency of sampling $\mathbf{z}$ from $p(\mathbf{z})$ and $q_\phi(\mathbf{z}|\mathbf{x}_{0:T})$. In the time-invariant version, we sample $\mathbf{z}$ once per video, whereas with the time-variant latent, sampling happens every frame. The main benefit of time-variant latent variable is better generalization beyond $T$, since the model does not have to encode all the events of the video in one vector $\mathbf{z}$. We provide an empirical comparison of these formulations in Section 5.2.

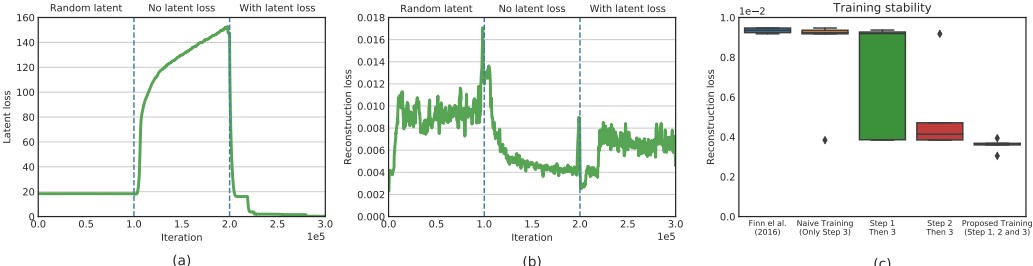

(a)  (b)  (c)

Figure 4: Three phases of training. In the first phase, the inference network is turned off and only the generative network is being trained, resulting in deterministic predictions. The inference network is used in the second phase without a KL-loss. The last phase includes $D_{KL}\big(q_\phi(\mathbf{z}|\mathbf{x}_{0:T})||p(\mathbf{z})\big)$ to enable accurate sampling latent from $p(\mathbf{z})$. *(a)* the KL-loss *(b)* the reconstruction loss *(c)* Training stability. This graph compares reconstruction loss at the end of five training sessions on the BAIR robot pushing dataset, with and without following all the steps of the training procedure. The proposed training is quite stable and results in lower error compared to naïve training.

In action-conditioned settings, we modify the generative model to be conditioned on action vector $\mathbf{a}_t$. This results in $p(\mathbf{z}_t)\prod_{t=c}^{T} p_\theta(\mathbf{x}_t|\mathbf{x}_{0:t-1}, \mathbf{z}_t, \mathbf{a}_t)$ as generative model while keeping the posterior approximation intact. Conditioning the outcome on actions can decrease future variability; however it will not eliminate it if the environment is inherently stochastic or the actions are ambiguous. In this case, the model is still capable of predicting stochastic outcomes in a narrower range of possibilities.

### 3.1 MODEL ARCHITECTURE

To model the approximated posterior $q_\phi(\mathbf{z}|\mathbf{x}_{0:T})$ we used a deep convolutional neural network as shown in the top row of Figure 3. Since we assumed a diagonal Gaussian distribution for $q_\phi(\mathbf{z}|\mathbf{x}_{0:T})$, this network outputs the mean $\mu_\phi(\mathbf{x}_{0:T})$ and standard deviation $\log \sigma_\phi(\mathbf{x}_{0:T})$ of the approximated posterior. Hence the entire inference network is convolutional, the predicted parameters are $8{\times}8$ single channel response maps. We assume each entry in this response maps is pairwise independent, forming the latent vector $\mathbf{z}$. The latent value is then sampled using Equation 1. As discussed before, this sampling happens every frame for time-varying latent, and once per video in time-invariant case.

For $p(\mathbf{x}_t|\mathbf{x}_{0:t-1}, \mathbf{z})$, we used the CDNA architecture proposed by Finn et al. (2016), which is a deterministic convolutional recurrent network that predicts the next frame $\mathbf{x}_t$ given the previous frame $\mathbf{x}_{t-1}$ and an optional action $\mathbf{a}_t$. This model constructs the next frames by predicting the motions of segments of the image (i.e., objects) and then merging these predictions via masking. Although this model directly outputs pixels, it is partially-appearance invariant and can generalize to unseen objects (Finn et al., 2016). To condition on the latent value, we modify the CDNA architecture by stacking $\mathbf{z}_t$ as an additional channel on tiled action $\mathbf{a}_t$.

### 3.2 TRAINING PROCEDURE

Our model can be trained end-to-end. However, our experiments show that naïve training usually results in the model ignoring the latent variables and converging to a suboptimal deterministic solution (Figure 4). Therefore, we train the model end-to-end in three phases, as follows:

1. **Training the generative network:** In this phase, the inference network has been disabled and the latent value $\mathbf{z}$ will be randomly sampled from $\mathcal{N}(\mathbf{0}, \mathbf{I})$. The intuition behind this phase is to train the generative model to predict the future frames deterministically (i.e. modeling $p_\theta(\mathbf{x}_t|\mathbf{x}_{0:t-1})$).

2. **Training the inference network:** In the second phase, the inference network is trained to estimate the approximate posterior $q_\phi(\mathbf{z}|\mathbf{x}_{0:T})$; however, the KL-loss is set to $0$. This means that the model can use the latent value without being penalized for diverging from $p(\mathbf{z})$. As seen in Figure 4, this phase results in very low reconstruction error, however it is not usable at the test time since $D_{KL}\big(q_\phi(\mathbf{z}|\mathbf{x}_{0:T})||p(\mathbf{z})\big) \gg 0$ and sampling $\mathbf{z}$ from the assumed prior will be inaccurate.

3. **Divergence reduction:** In the last phase, the KL-loss is added, resulting in a sudden drop of KL-divergence and an increase of reconstruction error. The reconstruction loss converging to a value lower than the first phase and KL-loss converging to zero are indicators of successful training. This means that $\mathbf{z}$ can be sampled from $p(\mathbf{z})$ at test time for effective stochastic prediction.

To gradually transition from the second phase to the third, we add a multiplier to KL-loss that is set to zero during the first two phases and then increased slowly in the last phase. This is similar to the $\beta$ hyper-parameter in Higgins et al. (2016) and Bowman et al. (2016) that is used to balance latent channel capacity and independence constraints with reconstruction accuracy.

We found that this training procedure is quite stable and the model almost always converges to the desired parameters. To demonstrate this stability, we trained the model with and without the proposed training procedure, five times each. Figure 4 shows the average and standard deviation of reconstruction loss at the end of these training sessions. Naïve training results in a slightly better error compared to Finn et al. (2016), but with high variance. When following the proposed training algorithm, the model consistently converges to a much lower reconstruction error.

## 4  STOCHASTIC MOVEMENT DATASET

To highlight the importance of stochasticity in video prediction, we created a toy video dataset with intentionally stochastic motion. Each video in this dataset is four frames long. The first frame contains a random shape (triangle, rectangle or circle) with random size and color, centered in the frame, which then randomly moves to one of the eight directions (up, down, left, right, up-left, up-right, down-left, down-right). Each frame is $64\times64\times3$ and the background is static gray. The main intuition behind this design is that, given only the first frame, a model can figure out the shape, color, and size of the moving object, but not its movement direction.

We train Finn et al. (2016) and SV2P to predict the future frames, given only the first frame. Figure 1 shows the video predictions from these two models. Since Finn et al. (2016) is a deterministic model with mean squared error as loss, it predicts the average of all possible outcomes, as expected. In contrast, SV2P predicts different possible futures for each sample of the latent variable $\mathbf{z} \sim \mathcal{N}(\mathbf{0}, \mathbf{I})$. In our experiments, all the videos predicted by SV2P are within the range of plausible futures (e.g. we never saw the shape moves in any direction other than the original eight). However, in some cases, SV2P still predicts the average of more than one future, as it can be seen in the first random sample of Figure 1c. The main reason for this problem seems to be overlapping posterior distributions in latent space which can cause some latent values (sampled from $p(\mathbf{z})$) to be ambiguous.

To demonstrate that the inference network is working properly and that the latent variable does indeed learn to store the information necessary for stochastic prediction (i.e., the direction of movement), we include predicted futures when $\mathbf{z} \sim q_\phi(\mathbf{x}_{0:T})$. By estimating the correct parameters of the latent distribution, using the inference network, the model always generates the right outcome. However, this cannot be used in practice, since the inference network requires access to all the frames, including the ones in the future. Instead, $\mathbf{z}$ will be sampled from assumed prior $p(\mathbf{z})$.

## 5  EXPERIMENTS

To evaluate SV2P, we test it on three real-world video datasets by comparing it to the CDNA model (Finn et al., 2016), as a deterministic baseline, as well as a baseline that outputs the last seen frame as the prediction. We compare SV2P with an auto-regressive stochastic model, video pixel networks (VPN) (Kalchbrenner et al., 2017). We use the parallel multi-resolution implementation of VPN from Reed et al. (2017), which is an order of magnitude faster than the original VPN, but still requires more than a minute to generate one second of $64\times64$ video. In all of these experiments, we plot the results of sampling the latent once per video (SV2P time-invariant latent) and once per frame (SV2P time-variant latent). We strongly encourage readers to view `https://goo.gl/iywUHc` for videos of the results which are more illustrative than printed frames.

### 5.1  DATASETS

We quantitatively and qualitatively evaluate SV2P on following real-world datasets:

- **BAIR robot pushing dataset** (Ebert et al., 2017): This dataset contains action-conditioned videos collected by a Sawyer robotic arm pushing a variety of objects. All of the videos in this datasets have similar table top settings with static background. Each video also has recorded actions taken by the robotic arm which correspond to the commanded gripper pose. An interesting property of this dataset is the fact that the arm movements are quite unpredictable in the absence of actions (compared to the robot pushing dataset (Finn et al., 2016) which the arm moves to the center of the bin). For this dataset, we train the models to predict the next ten frames given the first two, both in action-conditioned and action-free settings.

- **Human3.6M** (Ionescu et al., 2014): Humans and animals are one of the most interesting sources of stochasticity in natural videos, which behave in complex ways as a consequence of unpredictable intentions. To study human motion prediction, we use the Human3.6M dataset which consists of actors performing various actions in a room. We used the pre-processing and testing format of Finn et al. (2016): a 10 Hz frame rate and 10-frame prediction given the previous ten. The videos from this datasets contains various actions performed by humans (walking, talking on the phone, . . . ). Similar to Finn et al. (2016), we included videos from all the performed actions in training dataset while keeping all the videos from an specific actor out for testing.

- **Robotic pushing prediction** (Finn et al., 2016): We use the robot pushing prediction dataset to compare SV2P with another stochastic prediction method, video pixel networks (VPNs) (Kalchbrenner et al., 2017). VPNs demonstrated excellent results on this dataset in prior work, and therefore robot pushing dataset provides a strong point of comparison. However, in contrast to our method, VPNs do not include latent stochastic variables that represent random events, and rely on an expensive auto-regressive architecture. In this experiment, the models have been trained to predict the next ten frames, given the first two. Similar to BAIR robot pushing dataset, this dataset also contains actions taken by the robotic arm which are the pose of the commanded gripper.

## 5.2 QUANTITATIVE COMPARISON

In our quantitative evaluation, we aim to understand whether the range of possible futures captured by our stochastic model includes the true future. Models that are more stochastic do not necessarily score better on average standard metrics such as PSNR (Huynh-Thu & Ghanbari, 2008) and SSIM (Wang et al., 2004). However, if we are interested primarily in understanding whether the true outcome is within the set of predictions, we can instead evaluate the score of the **best** sample from multiple random priors. We argue that this is a better metric for stochastic models, since it allows us to understand if uncertain futures contain the true outcome. Figure 5 illustrates how this metric changes with different numbers of samples. By predicting more possible futures, the probability of predicting the true outcome increases, and therefore it is more likely to get a sample with higher PSNR compared to the ground truth. Of course, as with all video prediction metrics, it is imperfect, and is only suitable for understanding the performance of the model when combined with a visual examination of the qualitative results in Section 5.3.

To use this metric, we sample 100 latent values from prior $\mathbf{z} \sim \mathcal{N}(\mathbf{0}, \mathbf{I})$ and use them to predict 100 videos and show the result of the sample with

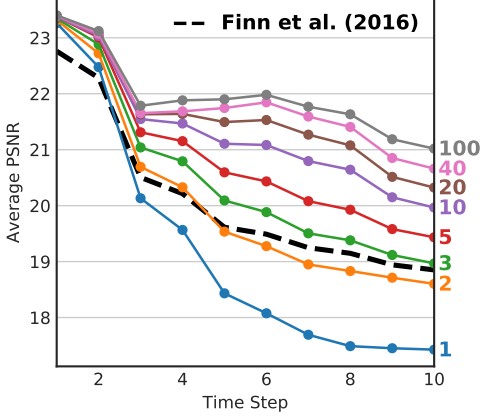

Figure 5: Stochasticity of SV2P predictions on the action-free BAIR dataset. Each line presents the sample with highest PSNR compared to ground truth, after multiple sampling. The number on the right indicates the number of random samples. As can be seen, SV2P predicts highly stochastic videos and, on average, only three samples is enough to predict outcomes with higher quality compared to Finn et al. (2016).

highest PSNR. For a fair comparison to VPN, we use the same best out of 100 samples for our stochastic baseline. Since even the fast implementation of VPN is quite slow, we limit the comparison with VPN to only last dataset with 256 test samples.

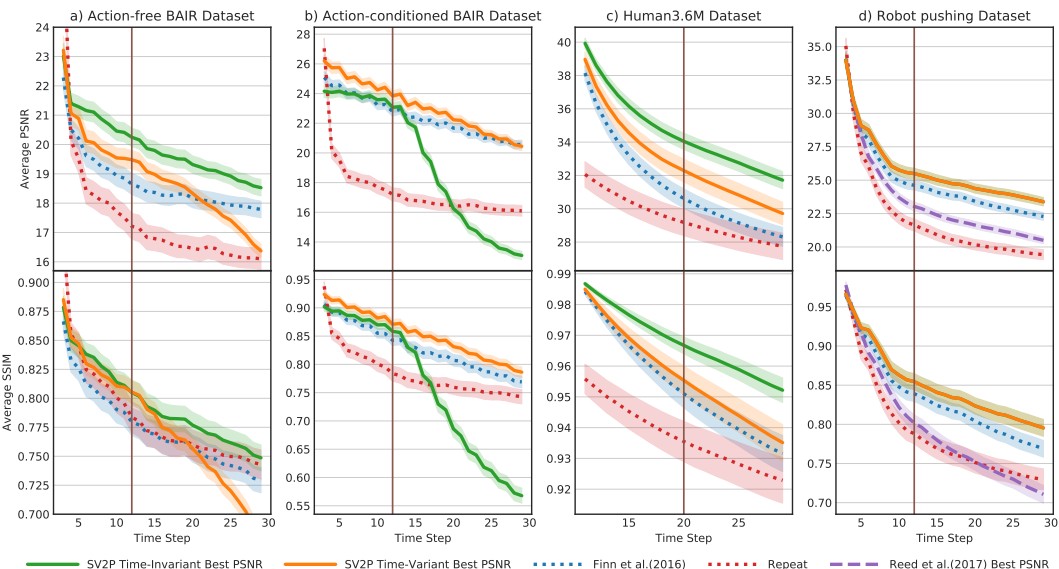

Figure 6: Quantitative comparison of the prediction methods. The stochastic models have been sampled 100 times and the results with the best PSNR have been displayed. For SV2P, we demonstrate the results of both time-variant and time-invariant latent sampling. *Repeat* shows the results of the lower bound prediction by repeating the last seen frame as the prediction. In the last column, we compare the results of video pixel networks (VPN). All the models, including Finn et al. (2016), have been trained up to the frame marked by vertical separator and the results beyond this line display their generalization. The plots are the average SSIM and PSNR over the test set and shadow is the 95% confidence interval. In all of these graphs, higher is better.

Figure 6 displays the quantitative comparison of the predictions on all of the datasets. In this graph, the top row is a PSNR comparison and the bottom row is SSIM, while each column represents a different dataset. To evaluate the generalization of the models beyond what they have been trained for, we generate more frames than what the models observed during training time. The length of the training sequences is marked by a vertical separator in all of the graphs, and the results beyond this line represent extrapolation to longer sequences.

Overall, SV2P with both time-variant and time-invariant latent sampling outperform all of the other baselines, by predicting higher quality videos with higher PSNR and SSIM. Time-varying latent sampling is more stable beyond the time horizon used during training (Figure 6b). One possible explanation for this behaviour is that the time-invariant latent has to include the information required for predicting all the frames and therefore, beyond training time, it collapses. This issue is mitigated by a time-variant latent variable which takes a different value at each time step. However, this stability is not always the case as it is more evident in late frames of Figure 6a.

One other interesting observation is that the time-invariant model outperforms the time-variant model in the Human3.6M dataset. In this dataset, the most important latent event – the action performed by the actor – is consistent across the whole video which is easier to capture using time-invariant latent.

## 5.3 QUALITATIVE COMPARISON

We can better understand the performance of the proposed model by visual examination of the qualitative results. We highlight some of the most important and observable differences in predictions by different models in Figures 8-11 [1]. In all of these figures, the x-axis is time (i.e., each row is one video). The first row is the ground truth video, and the second row is the result of Finn et al. (2016). The result of sampling the latent from approximated posterior is provided in the third row. For

---

[1]The videos of these experiments can be found at the project website (https://goo.gl/iywUHc).

stochastic methods, we show the best (highest PSNR) and worst (lowest PSNR) predictions out of 100 samples (as discussed in Section 5.2), as well as two random predicted videos from our model.

Figure 8 illustrates two examples from the BAIR robot pushing dataset in the action-free setting. As a consequence of the high stochasticity in the movement of the arm in absence of actions, Finn et al. (2016) only blurs the arm out, while SV2P predicts varied but coherent movements of the arm. Note that, although each predicted movements of the arm is random, it is still in the valid range of possible outcomes (i.e., there is no sudden jump of the arm nor random movement of the objects). The proposed model also learned how to move objects in cases where they have been pushed by the predicted movements of the arm, as can be seen in the zoomed images of both samples.

In the action-conditioned setting (Figure 9), the differences are more subtle: the range of possible outcomes is narrower, but we can still observe stochasticity in the behavior of the pushed objects. Interactions between the arm and objects are uncertain due to ambiguity in depth, friction, and mass, and SV2P is able to capture some of this variation. Since these variations are subtle and occupy a smaller part of the images, we illustrate this with zoomed insets in Figure 9. Some examples of varied object movements can be found in last three rows of right example of Figure 9. SV2P also generates sharper outputs, compared to Finn et al. (2016) as is evident in the left example of Figure 9.

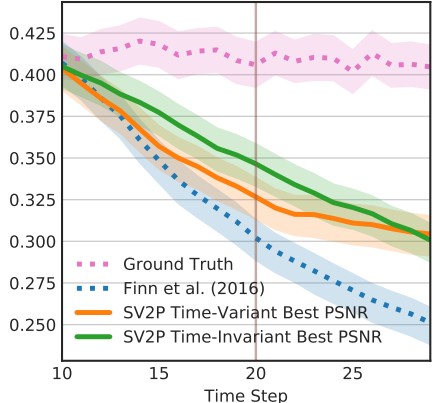

Figure 7: Quantitative comparison of the predicted frames on Human3.6M dataset using confidence of object detection as quality metric. The y-axis demonstrates the average confidence of Huang et al. (2016) in detecting humans in predicted frames. Based on this metric, SV2P predicts images with more meaningful semantics compared to to Finn et al. (2016).

Please note that the approximate posterior $q_\phi(\mathbf{z}|\mathbf{x}_{0:T})$ is still trained with the evidence lower bound (ELBO), which means that the posterior must compress the information of the future events. Perfect reconstruction of high-quality images from posterior distributions over latent states is an open problem, and the results in our experiments compare favorably to those typically observed even in single-image VAEs (e.g. see Xue et al. (2016)). This is why the model cannot reconstruct all the future frames perfectly, even though when latent values are sampled from $q_\phi(\mathbf{z}|\mathbf{x}_{0:T})$.

Figure 10 displays two examples from the Human3.6M dataset. In absence of actions, Finn et al. (2016) manages to separate the foreground from background, but cannot predict what happens next accurately. This results in distorted or blurred foregrounds. On the other hand, SV2P predicts a variety of different outcomes, and moves the actor accordingly. Note that PSNR and SSIM are measuring reconstruction loss with respect to the ground truth and they may not generally present a *better* prediction. For some applications, a prediction with lower PSNR/SSIM might have higher quality and be more interesting. A good example is the prediction with the worst PSNR in Figure 10-right, where the model predicts that the actor is spinning in his chair with relatively high quality. However, this output has the lowest PSNR compared to the ground truth.

However, pixel-wise metrics such as PSNR and SSIM may not be the best measures for semantic evaluation of predicted frames. Therefore, we use the confidence of an object detector to show the predicted frames contain useful semantic information. For this purpose, we use the open-sourced implementation of Huang et al. (2016) to compare the quality of predicted frames in Human3.6M dataset. As it can be seen in Figure 7, SV2P predicted frames which the human inside can be detected with higher confidence, compared to Finn et al. (2016).

Finally, Figure 11 demonstrates results on the Google robot pushing dataset. The qualitative and quantitative results in Figure 11 and 6 both indicate that SV2P produces substantially better predictions than VPNs. The quantitative results suggest that our best-of-100 metric is a reasonable measure of performance: the VPN predictions are more noisy, but simply increasing noise is not

sufficient to increase the quality of the best sample. The stochasticity in our predictions is more coherent, corresponding to differences in object or arm motion, while much of the stochasticity in the VPN predictions resembles noise in the image, as well as visible artifacts when predicting for substantially longer time horizons.

## 6 Conclusion

We proposed stochastic variational video prediction (SV2P), an approach for multi-step video prediction based on variational inference. Our primary contributions include an effective stochastic prediction method with latent variables, a network architecture that succeeds on natural videos, and a training procedure that provides for stable optimization. The source code for our method will be released upon acceptance. We evaluated our proposed method on three real-world datasets in action-conditioned and action-free settings, as well as one toy dataset which has been carefully designed to highlight the importance of the stochasticity in video prediction. Both qualitative and quantitative results indicate higher quality predictions compared to other deterministic and stochastic baselines.

SV2P can be expanded in numerous ways. First, the current inference network design is fully convolutional, which exposes multiple limitations, such as unmodeled spatial correlations between the latent variables. The model could be improved by incorporating the spatial correlation induced by the convolutions into the prior, using a learned structured prior in place of the standard spherical Gaussian. Time-variant posterior approximation to reflect the new information that is revealed as the video progresses, is another possible SV2P improvement. However, as discussed in Section 3, this requires incentivizing the inference network to incorporate the latent information at training time. This would allow time-variant latent distributions which is more aligned with generative neural models for time-series(Johnson et al., 2016; Gao et al., 2016; Krishnan et al., 2017).

Another exciting direction for future research would be to study how stochastic predictions can be used to act in the real world, producing model-based reinforcement learning methods that can execute risk-sensitive behaviors from raw image observations. Accounting for risk in this way could be especially important in safety-critical settings, such as robotics.

## Acknowledgement

The authors would like to thank Matt Johnson for providing feedback on an early draft of the paper, and Alex Lee for fixing bugs in the deterministic version of the model. This material is based upon work supported by the National Science Foundation under award no. 1725729 and was partially done while author was interning at Google Brain.

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

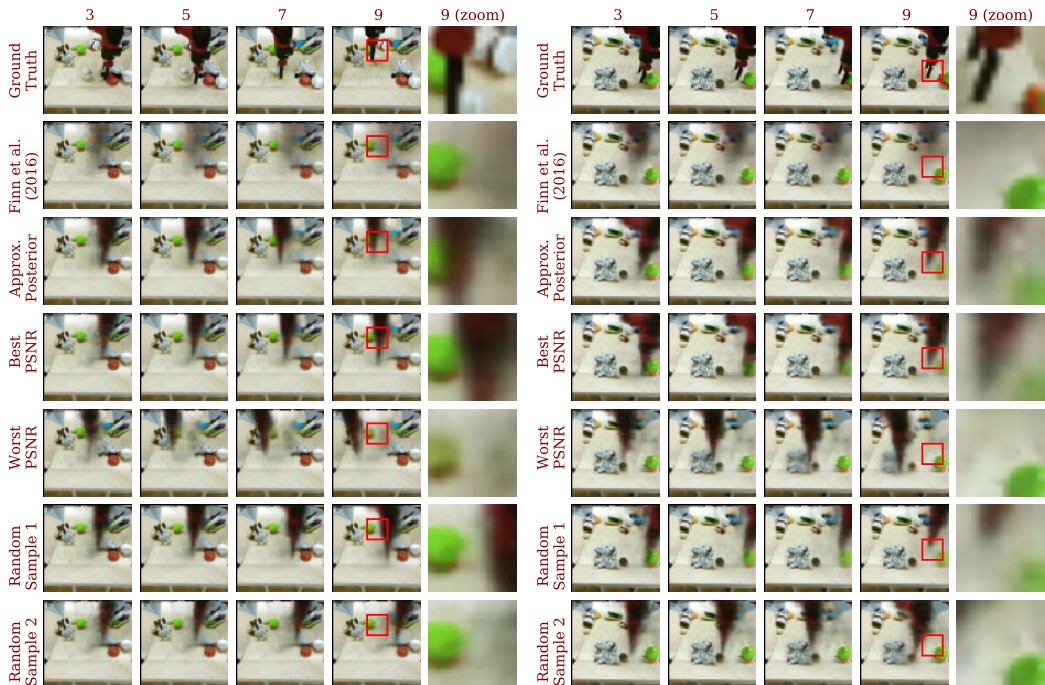

Figure 8: Comparing the results of SV2P with Finn et al. (2016) (second row) on action-free BAIR robot pushing dataset. Fourth and fifth rows are the predictions with minimum and maximum PSNR out of 100 random outputs with time-invariant latent sampling. The last two rows are random predicted outcomes. The numbers on top indicate the predicted frame number. In lack of actions and therefore high stochasticity, Finn et al. (2016) only blurs the robotic arm out while the proposed method predicts sharper frames on each sampling. SV2P also predicts the interaction dynamics between random movements of the arm and the objects.

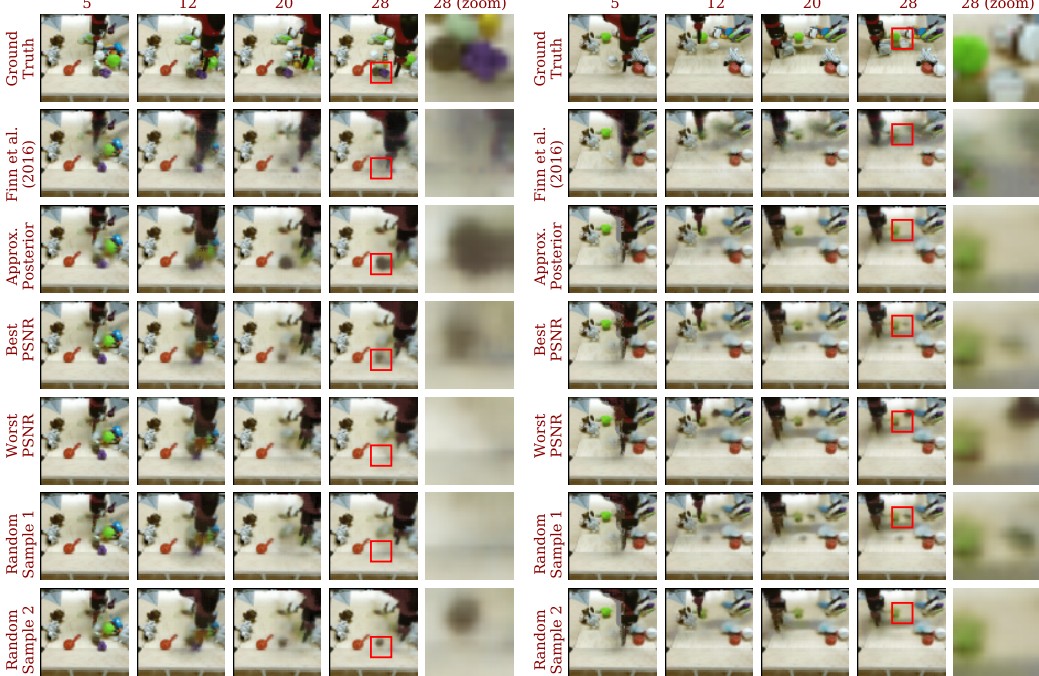

Figure 9: Similar comparison as Figure 8 this time action-conditioned with time-variant latent sampling. SV2P predicts sharper and slightly variant outcomes compared to Finn et al. (2016). This is mostly evident in zoomed in objects which have been pushed by the arm.

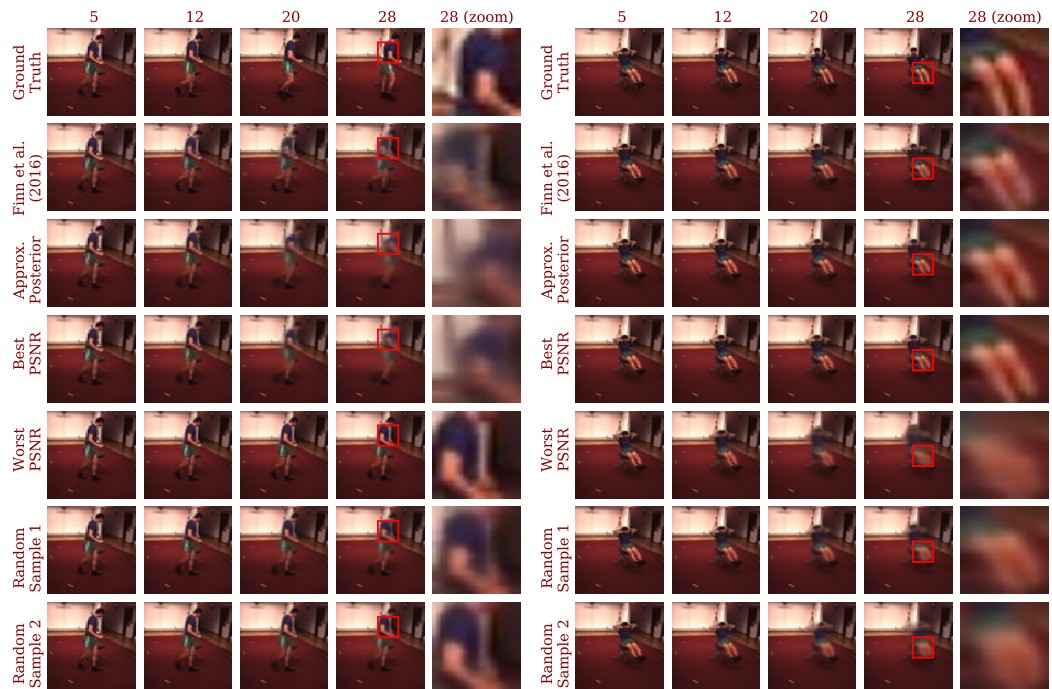

Figure 10: Prediction results on the action-free Human3.6M dataset. SV2P predicts a different outcome on each sampling given the latent. In the left example, the model predicts *walking* as well as *stopping* which result in different outputs in predicted future frames. Similarly, the right example demonstrates various outcomes including *spinning*.

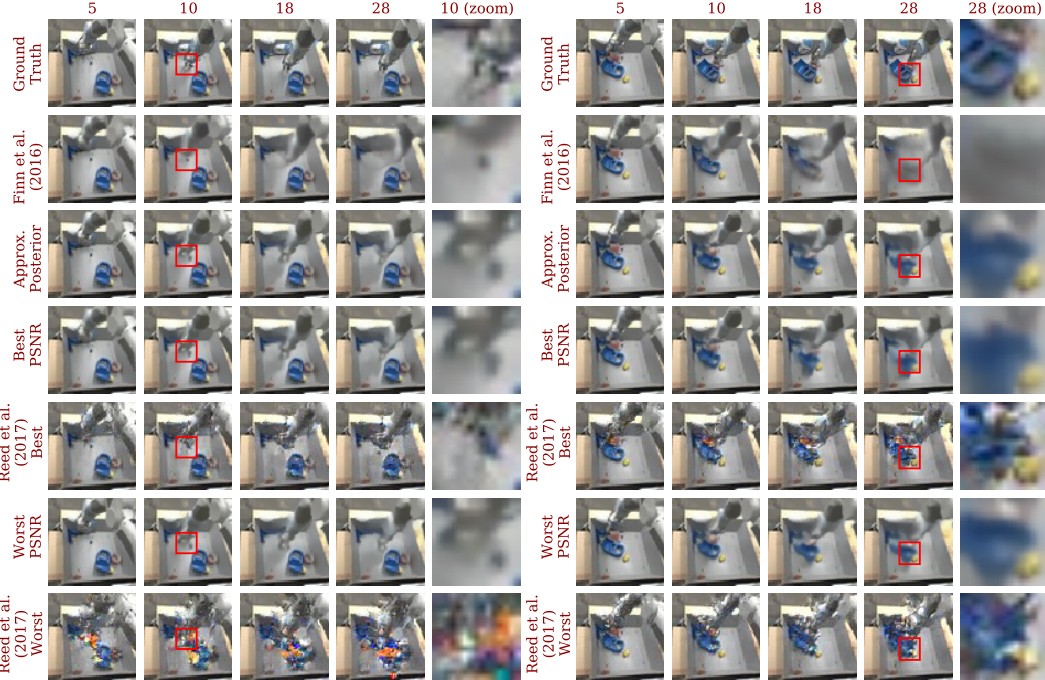

Figure 11: Comparing the results of video pixel networks (VPN) (Kalchbrenner et al., 2017; Reed et al., 2017) with SV2P on the robotic pushing dataset. We use the same best PSNR out of 100 random samples for both methods. Besides stochastic movements of the pushed objects, another source of stochasticity is the starting lag in movements of the robotic arm. SV2P generates sharper images compared to Finn et al. (2016) (notice the pushed objects in zoomed images) with less noise compared to Reed et al. (2017) (look at the accumulated noise in later frames).

Silvia Chiappa, Sébastien Racanière, Daan Wierstra, and Shakir Mohamed. Recurrent environment simulators. In *Proceedings of the International Conference on Learning Representations (ICLR)*, 2017.

Bert De Brabandere, Xu Jia, Tinne Tuytelaars, and Luc Van Gool. Dynamic filter networks. In *Neural Information Processing Systems (NIPS)*, 2016.

Emily Denton and Vighnesh Birodkar. Unsupervised learning of disentangled representations from video. *arXiv preprint arXiv:1705.10915*, 2017.

Frederik Ebert, Chelsea Finn, Alex X. Lee, and Sergey Levine. Self-Supervised Visual Planning with Temporal Skip Connections. *Conference on Robot Learning (CoRL)*, 2017.

Chelsea Finn and Sergey Levine. Deep visual foresight for planning robot motion. In *International Conference on Robotics and Automation (ICRA)*, 2017.

Chelsea Finn, Ian Goodfellow, and Sergey Levine. Unsupervised learning for physical interaction through video prediction. In *Advances in Neural Information Processing Systems*, 2016.

Katerina Fragkiadaki, Jonathan Huang, Alex Alemi, Sudheendra Vijayanarasimhan, Susanna Ricco, and Rahul Sukthankar. Motion prediction under multimodality with conditional stochastic networks. *CoRR*, abs/1705.02082, 2017.

Yuanjun Gao, Evan W Archer, Liam Paninski, and John P Cunningham. Linear dynamical neural population models through nonlinear embeddings. In *Advances in Neural Information Processing Systems*, pp. 163–171, 2016.

Ian Goodfellow. Nips 2016 tutorial: Generative adversarial networks. *arXiv preprint arXiv:1701.00160*, 2016.

Ian Goodfellow, Jean Pouget-Abadie, Mehdi Mirza, Bing Xu, David Warde-Farley, Sherjil Ozair, Aaron Courville, and Yoshua Bengio. Generative adversarial nets. In *Advances in neural information processing systems*, 2014.

Irina Higgins, Loic Matthey, Arka Pal, Christopher Burgess, Xavier Glorot, Matthew Botvinick, Shakir Mohamed, and Alexander Lerchner. beta-vae: Learning basic visual concepts with a constrained variational framework. *International Conference on Learning Representations (ICLR)*, 2016.

Jonathan Huang, Vivek Rathod, Chen Sun, Menglong Zhu, Anoop Korattikara, Alireza Fathi, Ian Fischer, Zbigniew Wojna, Yang Song, Sergio Guadarrama, et al. Speed/accuracy trade-offs for modern convolutional object detectors. *arXiv preprint arXiv:1611.10012*, 2016.

Quan Huynh-Thu and Mohammed Ghanbari. Scope of validity of psnr in image/video quality assessment. *Electronics letters*, 2008.

Catalin Ionescu, Dragos Papava, Vlad Olaru, and Cristian Sminchisescu. Human3. 6m: Large scale datasets and predictive methods for 3d human sensing in natural environments. *IEEE transactions on pattern analysis and machine intelligence*, 36(7), 2014.

Matthew Johnson, David K Duvenaud, Alex Wiltschko, Ryan P Adams, and Sandeep R Datta. Composing graphical models with neural networks for structured representations and fast inference. In *Advances in neural information processing systems*, pp. 2946–2954, 2016.

Nal Kalchbrenner, Aäron van den Oord, Karen Simonyan, Ivo Danihelka, Oriol Vinyals, Alex Graves, and Koray Kavukcuoglu. Video pixel networks. *International Conference on Machine Learning (ICML)*, 2017.

Diederik P Kingma and Max Welling. Auto-encoding variational bayes. *International Conference on Learning Representations (ICLR)*, 2014.

Rahul G Krishnan, Uri Shalit, and David Sontag. Structured inference networks for nonlinear state space models. In *AAAI*, pp. 2101–2109, 2017.

Yitong Li, Martin Renqiang Min, Dinghan Shen, David Carlson, and Lawrence Carin. Video generation from text. *arXiv preprint arXiv:1710.00421*, 2017.

Ziwei Liu, Raymond Yeh, Xiaoou Tang, Yiming Liu, and Aseem Agarwala. Video frame synthesis using deep voxel flow. *International Conference on Computer Vision (ICCV)*, 2017.

William Lotter, Gabriel Kreiman, and David Cox. Deep predictive coding networks for video prediction and unsupervised learning. *International Conference on Learning Representations (ICLR)*, 2017.

Michael Mathieu, Camille Couprie, and Yann LeCun. Deep multi-scale video prediction beyond mean square error. *International Conference on Learning Representations (ICLR)*, 2016.

Junhyuk Oh, Xiaoxiao Guo, Honglak Lee, Richard L Lewis, and Satinder Singh. Action-conditional video prediction using deep networks in atari games. In *Advances in Neural Information Processing Systems*, 2015.

MarcAurelio Ranzato, Arthur Szlam, Joan Bruna, Michael Mathieu, Ronan Collobert, and Sumit Chopra. Video (language) modeling: a baseline for generative models of natural videos. *arXiv preprint arXiv:1412.6604*, 2014.

Scott E. Reed, Aäron van den Oord, Nal Kalchbrenner, Sergio Gomez Colmenarejo, Ziyu Wang, Yutian Chen, Dan Belov, and Nando de Freitas. Parallel multiscale autoregressive density estimation. *International Conference on Machine Learning (ICML)*, 2017.

Rui Shu, James Brofos, Frank Zhang, Hung Hai Bui, Mohammad Ghavamzadeh, and Mykel Kochenderfer. Stochastic video prediction with conditional density estimation. In *ECCV Workshop on Action and Anticipation for Visual Learning*, 2016.

Nitish Srivastava, Elman Mansimov, and Ruslan Salakhudinov. Unsupervised learning of video representations using lstms. In *International Conference on Machine Learning*, 2015.

Sergey Tulyakov, Ming-Yu Liu, Xiaodong Yang, and Jan Kautz. Mocogan: Decomposing motion and content for video generation. *arXiv preprint arXiv:1707.04993*, 2017.

Carl Vondrick and Antonio Torralba. Generating the future with adversarial transformers. In *Computer Vision and Pattern Recognition (CVPR)*, 2017.

Carl Vondrick, Hamed Pirsiavash, and Antonio Torralba. Anticipating the future by watching unlabeled video. *arXiv preprint arXiv:1504.08023*, 2015.

Carl Vondrick, Hamed Pirsiavash, and Antonio Torralba. Generating videos with scene dynamics. In *Advances In Neural Information Processing Systems*, 2016.

Jacob Walker, Carl Doersch, Abhinav Gupta, and Martial Hebert. An uncertain future: Forecasting from static images using variational autoencoders. In *European Conference on Computer Vision*, pp. 835–851. Springer, 2016.

Zhou Wang, Alan C Bovik, Hamid R Sheikh, and Eero P Simoncelli. Image quality assessment: from error visibility to structural similarity. *IEEE transactions on image processing*, 2004.

SHI Xingjian, Zhourong Chen, Hao Wang, Dit-Yan Yeung, Wai-Kin Wong, and Wang-chun Woo. Convolutional lstm network: A machine learning approach for precipitation nowcasting. In *Advances in Neural Information Processing Systems*, 2015.

Tianfan Xue, Jiajun Wu, Katherine Bouman, and Bill Freeman. Visual dynamics: Probabilistic future frame synthesis via cross convolutional networks. In *Advances in Neural Information Processing Systems*, 2016.

Jun-Yan Zhu, Taesung Park, Phillip Isola, and Alexei A. Efros. Unpaired image-to-image translation using cycle-consistent adversarial networks. *International Conference on Computer Vision (ICCV)*, 2017.

## A    TRAINING DETAILS

Figure 3 contains details of the network architectures used as generative and inference models. In all of the experiments we used the same set of hyper-parameters which can be found in Table 1.

Table 1: Hyper-parameters used for experiments.

| **Generative Network** | |
| --- | --- |
| model type | CDNA |
| batch size | 16 |
| learning rate | 0.001 |
| scheduled sampling ($k$) | 900.0 |
| # of masks | 10 |
| # of iterations | 200000 |
| **Inference Network** | |
| latent minimum $\sigma$ | -5.0 |
| starting $\beta$ | 0.0 |
| final $\beta$ | 0.001 |
| # of latent channels | 1 |
| # step 1 iterations | 50000 |
| # step 2 iterations | 50000 |
| # step 3 iterations | 100000 |
| **Optimization** | |
| Method | ADAM |
| $\beta 1$ | 0.9 |
| $\beta 2$ | 0.999 |
| $\epsilon$ | 1e-8 |

In the first step of training, we disable the inference network and instead sample latent values from $\mathcal{N}(\mathbf{0}, \mathbf{I})$. In step 2, the latent values will be sampled from the approximated posterior $q_\phi(\mathbf{z}|\mathbf{x}_{0:T}) = \mathcal{N}(\mu(\mathbf{x}_{0:T}), \sigma(\mathbf{x}_{0:T}))$. Please note that the inference network approximates $\log(\sigma)$ instead of $\sigma$ for numerical stability. To gradually switch from Step 2 of training procedure to Step 3, we increase $\beta$ linearly from its starting value to its end value over the length of training.

