# OpenReview forum: "Stochastic Variational Video Prediction"
_ICLR.cc/2018/Conference — Accept (Poster)_

### Official Review · AnonReviewer3 · 2017-11-12
**Suggest an accept but it requires further revision**

**Rating:** 7
**Confidence:** 4

**Review:**

Quality: above threshold
Clarity: above threshold, but experiment details are missing.
Originality: slightly above threshold.
Significance: above threshold

Pros:

This paper proposes a stochastic variational video prediction model. It can be used for prediction in optionally available external action cases. The inference network is a convolution net and the generative network is using a previously structure with minor modification. The result shows its ability to sample future frames and outperforms with methods in qualitative and quantitive metrics.

Cons:

1. It is a nice idea and it seems to perform well in practice, but are there careful experiments justifying the 3-stage training scheme? For example, compared with other schemes like alternating between 3 stages, dynamically soft weighting terms.

2. It is briefly mentioned in the context, but has there any attempt towards incorporating previous frames context for z, instead of sampling from prior? This piece seems much important in the scenarios which this paper covers.

3. No details about training (training data size, batches, optimization) are provided in the relevant section, which greatly reduces the reproducibility and understanding of the proposed method. For example, it is not clear whether the model can generative samples that are not previously seen in the training set. It is strongly suggested training details be provided.

4. Minor, If I understand correctly, in equation in the last paragraph above 3.1,  z instead of z_t

---

> ### Author Response · Authors · 2017-12-27
> **Updated paper + addressing comments.**
>
> Thank you for your great comments. comments and constructive criticism. We updated the paper to address all of your comments. Please let us know if you have any more suggestions or comments. Thanks!
>
>
> - “No details about training (training data size, batches, optimization) are provided in the relevant section”
>
> Thank you for your great comment. To further investigate the effect of our proposed training method, we conducted more experiments by alternating between different steps of suggested method. The updated Figure 4c reflects the results of this experiments. As it can be seen in this graph, the suggested steps help with both stability and convergence of the model.
>
> We also provided details of the training method in Appendix A to address your comment regarding using soft-terms as well as reproducibility. We will also release the code after acceptance.
>
>
> - “It is briefly mentioned in the context, but has there any attempt towards incorporating previous frames context for z, instead of sampling from prior? This piece seems much important in the scenarios which this paper covers.“
>
> This is one of the future work directions mentioned in the conclusion section. We’ve expanded this discussion in the conclusion a bit to address this better.
>
>
> - “Minor, If I understand correctly, in equation in the last paragraph above 3.1, z instead of z_t”
>
> Thank you for the detailed comment. We fixed the typo.

---

### Official Review · AnonReviewer1 · 2017-11-25

**Rating:** 7
**Confidence:** 5

**Review:**

1) Summary
This paper proposed a new method for predicting multiple future frames in videos. A new formulation is proposed where the frames’ inherent noise is modeled separate from the uncertainty of the future. This separation allows for directly modeling the stochasticity in the sequence through a random variable z ~ p(z)  where the posterior  q(z | past and future frames) is approximated by a neural network, and as a result, sampling of a random future is possible through sampling from the prior p(z) during testing. The random variable z can be modeled in a time-variant and time-invariant way. Additionally, this paper proposes a training procedure to prevent their method from ignoring the stochastic phenomena modeled by z. In the experimental section, the authors highlight the advantages of their method in 1) a synthetic dataset of shapes meant to clearly show the stochasticity in the prediction, 2) two robotic arm datasets for video prediction given and not given actions, and 3) A challenging human action dataset in which they perform future prediction only given previous frames.



2) Pros:
+ Novel/Sound future frame prediction formulation and training for modeling the stochasticity of future prediction.
+ Experiments on the synthetic shapes and robotic arm datasets highlight the proposed method’s power of multiple future frame prediction possible.
+ Good analysis on the number of samples improving the chance of outputting the correct future, the modeling power of the posterior for reconstructing the future, and a wide variety of qualitative examples.
+ Work is significant for the problem of modeling the stochastic nature of future frame prediction in videos.




3) Cons:
Approximate posterior in non-synthetic datasets:
The variable z seems to not be modeling the future very well. In the robot arm qualitative experiments, the robot motion is well modeled, however, the background is not. Given that for the approximate posterior computation the entire sequence is given (e.g. reconstruction is performed), I would expect the background motion to also be modeled well. This issue is more evident in the Human 3.6M experiments, as it seems to output blurriness regardless of the true future being observed. This problem may mean the method is failing to model a large variety of objects and clearly works for the robotic arm because a very similar large shape (e.g. robot arm) is seen in the training data. Do you have any comments on this?



Finn et al 2016 PNSR performance on Human 3.6M:
Is the same exact data, pre-processing, training, and architecture being utilized? In her paper, the PSNR for the first timestep on Human 3.6M is about 41 (maybe 42?)  while in this paper it is 38.



Additional evaluation on Human 3.6M:
PSNR is not a good evaluation metric for frame prediction as it is biased towards blurriness, and also SSIM does not give us an objective evaluation in the sense of semantic quality of predicted frames. It would be good if the authors present additional quantitative evaluation to show that the predicted frames contain useful semantic information [1, 2, 3, 4]. For example, evaluating the predicted frames for the Human 3.6M dataset to see if the human is still detectable in the image or if the expected action is being predicted could be useful to verify that the predicted frames contain the expected meaningful information compared to the baselines.



Additional comments:
Are all 15 actions being used for the Human 3.6M experiments? If so, the fact of the time-invariant model performs better than the time-variant one may not be the consistent action being performed (last sentence of 5.2). The motion performed by the actors in each action highly overlaps (talking on the phone action may go from sitting to walking a little to sitting again, and so on). Unless actions such as walking and discussion were only used, it is unlikely the time-invariant z is performing better because of consistent action. Do you have any comments on this?



4) Conclusion
This paper proposes an interesting novel approach for predicting multiple futures in videos, however, the results are not fully convincing in all datasets. If the authors can provide additional quantitative evaluation besides PSNR and SSIM (e.g. evaluation on semantic quality), and also address the comments above, the current score will improve.



References:
[1] Emily Denton and Vighnesh Birodkar. Unsupervised Learning of Disentangled Representations from Video. In NIPS, 2017.
[2] Ruben Villegas, Jimei Yang, Yuliang Zou, Sungryull Sohn, Xunyu Lin, and Honglak Lee. Learning to generate long-term future via hierarchical prediction. In ICML, 2017.
[3] Tero Karras, Timo Aila, Samuli Laine, and Jaakko Lehtinen. Progressive Growing of GANs for Improved Quality, Stability, and Variation. arXiv preprint arXiv:1710.10196, 2017.
[4] Tim Salimans, Ian Goodfellow, Wojciech Zaremba, Vicki Cheung, Alec Radford, and Xi Chen. Improved Techniques for Training GANs. In NIPS, 2017.


Revised review:
Given the authors' thorough answers to my concerns, I have decided to change my score. I would like to thank the authors for a very nice paper that will definitely help the community towards developing better video prediction algorithms that can now predict multiple futures.

---

> ### Author Response · Authors · 2017-12-27
> **Updated paper + addressing comments.**
>
> Thank you for your insightful comments and suggestions. We have addressed most of your concerns. Please see our responses below and let us know if you have any further comments on the paper. Thanks!
>
> - "Additional evaluation on Human 3.6M: PSNR is not a good evaluation metric for frame prediction"
>
> Thank you for this suggestion. We’ve updated the paper (please look at Figure 7 and 6th paragraph of 5.3) to address your comment. In order to investigate the quality difference between SV2P predicted frames and “Finn et al (2016)”, we performed a new experiment in which we used the open sourced version of the object detector from “Huang et al. (2016)”:
> https://github.com/tensorflow/models/blob/master/research/object_detection/models/ssd_mobilenet_v1_feature_extractor.py
> to detect the humans inside the predicted frames. We used the confidence of this detection as an additional metric to evaluate the difference between different methods. The results of this comparison which shows higher quality for SV2P can be found in (newly added) Figure 7.
>
>
> - "Are all 15 actions being used for the Human 3.6M experiments?"
>
> We’ve updated the 2nd bullet point in 5.1 to clear this up in the paper. Yes, we are using all the actions. In regard to changing actions: since the videos are relatively short (about 20 frames), there aren't any videos where the actor changes the behavior in the middle. That said, the identity of the behavior is not the only source of stochasticity, since even within a single action (e.g., walking), the actor might choose to walk at different speeds and in different directions.
>
>
> - "I would expect the background motion to also be modeled well.”
>
> We've added a discussion of this in Section 5.3 (paragraph 4). Note that the approximate posterior over z is still trained with the ELBO, which means that it must compress the information in future events. Perfect reconstruction of high-quality images from posterior distributions over latent states is an open problem, and the results in our experiments compare favorably to those typically observed even in single-image VAEs (e.g. see Xue et al. (2016))
>
>
> - "Finn et al 2016 PNSR performance on Human 3.6M: In her paper, the PSNR for the first timestep on Human 3.6M is about 41 (maybe 42?) while in this paper it is 38"
>
> For “Finn et al. (2016)”, we used the open-source version of the code here:
> https://github.com/tensorflow/models/tree/master/research/video_prediction
> which is a reimplementation of the models used in the Finn et al. ‘16 paper. We are not exactly sure where the discrepancy is coming from. However, we would like to point out that whatever issue resulted in slightly slower PSNR for the deterministic model would have affected our model as well, since we used the same code for the base model. Hence, the comparison is still valid.

---

### Official Review · AnonReviewer2 · 2017-11-26
**Convincing demonstration of stochastic video predictions on real data**

**Rating:** 7
**Confidence:** 4

**Review:**

The submission presents a method or video prediction from single (or multiple) frames, which is capable of producing stochastic predictions by means of training a variational encoder-decoder model. Stochastic video prediction is a (still) somewhat under-researched direction, due to its inherent difficulty.

The method can take on several variants: time-invariant [latent variable] vs. time-variant, or action-conditioned vs unconditioned. The generative part of the method is mostly borrowed from Finn et al. (2016). Figure 1 clearly motivates the problem. The method itself is fairly clearly described in Section 3; in particular, it is clear why conditioning on all frames during training is helpful. As a small remark, however, it remains unclear what the action vector a_t is comprised of, also in the experiments.

The experimental results are good-looking, especially when looking at the provided web site images.
The main goal of the quantitative comparison results (Section 5.2) is to determine whether the true future is among the generated futures. While this is important, a question that remains un-discussed is whether all generated stochastic samples are from realistic futures. The employed metrics (best PSNR/SSIM among multiple samples) can only capture the former, and are also pixel-based, not perceptual.

The quantitative comparisons are mostly convincing, but Figure 6 needs some further clarification. It is mentioned in the text that "time-varying latent sampling is more stable beyond the time horizon used during training". While true for Figure 6b), this statement is contradicted by both Figure 6a) and 6c), and Figure 6d) seems to be missing the time-invariant version completely (or it overlaps exactly, which would also need explanation). As such, I'm not completely clear on whether the time variant or invariant version is the stronger performer.

The qualitative comparisons (Section 5.3) are difficult to assess in the printed material, or even on-screen. The animated images on the web site provide a much better impression of the true capabilities, and I find them convincing.

The experiments only compare to Reed et al. (2017)/Kalchbrenner et al. (2017), with Finn at al. (2016) as a non-stochastic baseline, but no comparisons to, e.g., Vondrick et al. (2016) are given. Stochastic prediction with generative adversarial networks is a bit dismissed in Section 2 with a mention of the mode-collapse problem.

Overall the submission makes a significant enough contribution by demonstrating a (mostly) working stochastic prediction method on real data.

---

> ### Author Response · Authors · 2017-12-27
> **Updated paper + addressing comments.**
>
> Thank you for insightful comments and constructive criticism. We updated the paper to address all of your comments. Please let us know if you have any more suggestions or comments. Thanks!
>
> - "As a small remark, however, it remains unclear what the action vector a_t is comprised of, also in the experiments."
>
> Thank you for the good point. We’ve updated the paper to clarify what the actions are for each dataset. Please check 5.1 for more clarification.
>
>
> - "a question that remains un-discussed is whether all generated stochastic samples are from realistic futures"
>
> We’ve updated the paper to clarify this issue. Please look at the 2nd paragraph of Section 4 for updates. In short, as we observed empirically from the predicted videos, the output videos are within the realistic possibilities. However, in some cases, the predicted frames are not realistic and are averaging more than one future (e.g. first random sample in Figure 1-C).
>
>
> - "The employed metrics (best PSNR/SSIM among multiple samples) can only capture the former, and are also pixel-based, not perceptual."
>
> Thank you for the great comment. We’ve updated the paper (please look at Figure 7 and 6th paragraph of 5.3) to address your comment. In order to investigate the quality difference between SV2P predicted frames and “Finn et al (2016)”, we performed a new experiment in which we used the open sourced version of the object detector from “Huang et al. (2016)”:
> https://github.com/tensorflow/models/blob/master/research/object_detection/models/ssd_mobilenet_v1_feature_extractor.py
> to detect the humans inside the predicted frames. We used the confidence of this detection as an additional metric to evaluate the difference between different methods. The results of this comparison which shows higher quality for SV2P can be found in (newly added) Figure 7.
>
>
> - "time-varying latent sampling is more stable beyond the time horizon used during training". While true for Figure 6b), this statement is contradicted by both Figure 6a) and 6c)."
>
> Thank you for the great question. We’ve updated the paper (last two paragraphs of 5.2) to include your observation. Please note that our original claim was that the time-variant latent seems to be more “stable” beyond the time horizon used during training (which is highly evident in Figure 6b). And we are NOT claiming that time-variant latent generates “higher quality” results. However,  we agree that this stability is not always the case as it is more evident in late frames of Figure 6a.

---

### Public Comment · (anonymous) · 2017-11-20
**Prior Work**

The authors make the following claim:

"We believe, our approach is the first latent variable model to successfully demonstrate stochastic multi-frame video prediction on real world datasets."

However, variational methods have been used before to forecast multiple frames in static images (An Uncertain Future:
Forecasting from Static Images using Variational Autoencoders, Walker et al., ECCV 2016). In this ECCV paper, the output space
are dense pixel trajectories instead of direct pixels, but the model is trained on realistic videos of human activities. What distinguishes this proposed approach from prior work? The paper has been cited in references of other papers cited by the authors.

---

> ### Author Response · Authors · 2017-11-21
> **re: prior work**
>
> We apologize for missing that highly-relevant reference. We will include a reference in the next revision.
>
> Note that Walker et al. '16 does not predict video frames; thus, we cannot compare to the approach.  Unlike Walker et al. '16, our work does not require optical flow supervision nor an optical flow solver, which tend to not work consistently on real videos (as optical flow is an open research problem [1,2,3]). Our method only uses raw videos. Furthermore, we show that a CVAE trained from scratch does not work consistently, and propose a pre-training scheme which, in our experiments, consistently finds a good solution.
>
> [1] https://arxiv.org/abs/1705.01352
> [2] https://arxiv.org/abs/1612.01925
> [3] https://arxiv.org/abs/1604.01827

---

### Public Comment · (anonymous) · 2017-12-27
**Training & Model Questions**

Nice work! A couple of questions about the architecture and training procedure.

- For the time-variant latent variable case, the paper says that the inference model is q_phi(z_t | x_{0:T}). I want to make sure I'm understanding the time-variant latent setup right - is the inference process exactly the same at all time steps t? This seems a bit puzzling to me. Why does it help to sample latents multiple times if the inference procedure is identical at all time steps? Is it simply because you get extra bits of stochasticity?

- The plot in figure 4a shows the KL loss going to 0. This seems odd to me, because the KL term in a VAE usually roughly corresponds with the diversity of the samples. If it's close to 0, then the information passing through the posterior is close to 0, isn't it? Furthermore, in the third phase, where you increase the multiplicative constant associated with the KL (beta), it seems surprising that you only need to increase it to 0.001. If beta is only 0.001, shouldn't the KL be fairly high?

- In the inference network (top), could you please be more specific about how you transform the Tx64x64x3 tensor into a 32x32x32 tensor (combining the dimensions across time)? Thanks!

- SV2P uses an unconditional prior when generating samples. The top of page 4 (caption for Figure 3) gives an argument for this choice. However, I don't buy the argument for why the "filtering process at training time" (as far as I understand, basically conditioning the prior/posterior on the context) won't work. In particular, the "extra" information should come from the frames after the context/seed frames to the end of the video. Could you please explain what kind of experiments you ran (informally) to check that this doesn't work better?

---

> ### Author Response · Authors · 2017-12-31
> **Addressing comment.**
>
> Thank you for the great comment. We address your old version (since it had some great questions) as well as the updated version. Please let us know if we missed a question and/or if you have more questions/comments.
>
> - Why does it help to sample latents multiple times if the inference procedure is identical at all time steps? Is it simply because you get extra bits of stochasticity?
>
> Thank you for the great question. Please note that we are not claiming the time-invariant latent predicts higher quality images. The claim is that it is more stable beyond training time horizon (look at Figure 6b). It is best if we answer this questions intuitively with an example. Think about a simple shape which moves randomly and changes its direction in each time step (e.g. brownian motion). A time-invariant latent should encode the info about *all* of the time steps and the generative model should learn how to decode all of this information, step by step, and therefore it runs out of *information* after all the time-steps which causes the collapse after the training time horizon. However, a time-variant latent only includes information about the *current* time frames and stays stable after any time horizon. However, this pushes the complexity to posterior approximation since it should *find* a distribution aligned with what is happening at training time. That is why the result are not that different (other plots of Figure 6). This can be improved by conditioning the prior or posterior on input (which we are currently working on) or other techniques such as backproping through the best out of multiple samples (e.g. look at Fragkiadaki et al. (2017)).
>
> - The plot in figure 4a shows the KL loss going to 0. This seems odd to me, because the KL term in a VAE usually roughly corresponds with the diversity of the samples. If it's close to 0, then the information passing through the posterior is close to 0, isn't it?
>
> It does not go to 0 but converges to a small number, usually 3 to 5 (please note that y-axis has a very large scale). Intuitively, the key is to keep the divergence small enough so sampling from prior at test time still makes sense, and big enough so there is enough information to train the generative network. In our experiments we found this magical number to be around 3 to 5.
>
> - In the third phase, where you increase the KL, do you just increase the KL to 1? Or, like in Higgins et al (2016), do you tune the multiplicative constant for the KL term? Did you try any (informal) experiments with other types of pre-training. What did/did not seem to work well?
>
> Please check (the newly added) Appendix. In the current setting we do not increase KL to 1 but increase it to 1e-3. Regarding the training we also included more information in the updated Figure 4. We tried variations of the proposed training mechanism to see how it affects the training. Besides that (informally) we tried different approaches for KL annealing. Since the explained training mechanism is practical enough, we stopped exploring more.
>
> - In your case, I don't think the latent will capture the type of objects in the scene. Otherwise, the latents could "conflict" with the context.
>
> Great question! We indeed observed the conflict that you mentioned while we were developing the model. e.g. a green circle was being morphed into a red triangle! However, there are is a key remark which prevents this from happening in the current architecture and it’s the reconstruction loss. Intuitively, in the training time, the posterior should encode the information into a distribution in which *any* sample from it results to a correct answer to minimizes the loss. Therefore, first, it avoids adding any not necessary information which is accessible during generative (e.g. the shape and color). Second, it encodes all the required info for a correct prediction (e.g. movement). Therefore, as you mentioned, the latent values include only the *movement* info and not the context. This contains more information compared to random bits though.
>
> - In the inference network (top), could you please be more specific about how you transform the Tx64x64x3 tensor into a 32x32x32 tensor (combining the dimensions across time)? Thanks!
>
> Yes, we combine the time dimension and stride of 2 downsamples to 32x32

---

### Public Comment · (anonymous) · 2018-01-08
**Baselines**

Interesting work. A couple of questions about the comparison with Video Pixel Networks (VPN):

- Did you tune the parameters in the VPN model for your specific datasets? Did you try a similar number of hyperparameter combinations for VPN and SV2P?

- Do you have any metrics to suggest that the comparisons between SV2P and VPN are fair? For example, did you see better test set generalization (given the same training error) in both models. Or how many parameters are in the VPN architecture vs SV2P? I guess it's natural to ask if VPN could achieve the same test error if we simply scale up the model.

Its fine if you don't, I'm just wondering. Thanks!

---

> ### Author Response · Authors · 2018-01-20
> **Thank you for the questions.**
>
>
> For comparison with VPN, we did NOT train any model. Instead, the authors of Reed et al. 2017 provided their trained model which we used for evaluation.
>
> In terms of numbers, the model from Reed et al. 2017 has 119,538,432 while our model has 8,378,497. Hopefully this helps to get a better understanding of the generalizations.

---

### Decision · Program_Chairs · 2018-01-29
**ICLR 2018 Conference Acceptance Decision**

**Decision:**

Accept (Poster)

**Comment:**

Not quite enough for an oral but a very solid poster.